# Endothelial Contribution to Warfarin-Induced Arterial Media Calcification in Mice

**DOI:** 10.3390/ijms222111615

**Published:** 2021-10-27

**Authors:** Geoffrey Van den Bergh, Sofie De Moudt, Astrid Van den Branden, Ellen Neven, Hanne Leysen, Stuart Maudsley, Guido R. Y. De Meyer, Patrick D’Haese, Anja Verhulst

**Affiliations:** 1Laboratory of Pathophysiology, University of Antwerp, 2610 Wilrijk, Belgium; geoffrey.vandenbergh@uantwerpen.be (G.V.d.B.); astrid.vandenbranden@student.uantwerpen.be (A.V.d.B.); ellenneven@hotmail.com (E.N.); Patrick.dhaese@uantwerpen.be (P.D.); 2Laboratory of Physiopharmacology, University of Antwerp, 2610 Wilrijk, Belgium; sofie.demoudt@uantwerpen.be (S.D.M.); guido.demeyer@uantwerpen.be (G.R.Y.D.M.); 3Receptor Biology Laboratory, University of Antwerp, 2610 Wilrijk, Belgium; hanne.leysen@uantwerpen.be (H.L.); stuart.maudsley@uantwerpen.be (S.M.)

**Keywords:** vascular calcification, endothelial cells, vascular smooth muscle cells, nitric oxide, organ baths

## Abstract

Arterial media calcification (AMC) is predominantly regulated by vascular smooth muscle cells (VSMCs), which transdifferentiate into pro-calcifying cells. In contrast, there is little evidence for endothelial cells playing a role in the disease. The current study investigates cellular functioning and molecular pathways underlying AMC, respectively by, an ex vivo isometric organ bath set-up to explore the interaction between VSMCs and ECs and quantitative proteomics followed by functional pathway interpretation. AMC development, which was induced in mice by dietary warfarin administration, was proved by positive Von Kossa staining and a significantly increased calcium content in the aorta compared to that of control mice. The ex vivo organ bath set-up showed calcified aortic segments to be significantly more sensitive to phenylephrine induced contraction, compared to control segments. This, together with the fact that calcified segments as compared to control segments, showed a significantly smaller contraction in the absence of extracellular calcium, argues for a reduced basal NO production in the calcified segments. Moreover, proteomic data revealed a reduced eNOS activation to be part of the vascular calcification process. In summary, this study identifies a poor endothelial function, next to classic pro-calcifying stimuli, as a possible initiator of arterial calcification.

## 1. Introduction

Arterial calcification can be roughly defined as the pathological deposition of calcium-phosphate crystals in the arterial wall [1]. While the vascular intimal layer mainly is comprised of endothelial cells (ECs), the medial layer is primarily composed of vascular smooth muscle cells (VSMCs), reinforced with elastin and collagen fibrils [2]. Arterial calcification has been classified based on the site of crystal deposition. As such, this process can occur both in the intimal layer and medial layer of the arteries. The former is associated with atherosclerosis, and the latter, also known as arterial media calcification (AMC) is a consequence of normal vascular aging. AMC is highly prevalent in the elderly and is accelerated in patients with underlying chronic kidney disease (CKD), osteoporosis, hypertension and Type 2 diabetes mellitus [3].

AMC is an independent risk factor for cardiovascular morbidity and mortality, as the presence of calcification in the vessel wall is associated with a threefold higher risk of mortality and deleterious cardiovascular events [4]. Moreover, early-stage CKD patients are more likely to die from cardiovascular disease than to further develop end-stage renal disease [5]. As AMC itself, pathologies associated with arterial calcification, including diabetes mellitus and CKD are strongly associated with aging. Currently, >20% of the European population is older than 65 years and respectively 9.1–13.4% [6,7] and 12.3% [8] of the population suffers from CKD and/or diabetes mellitus. These data therefore suggest that a growing impact of AMC upon healthcare in future years is highly likely.

To date, there are no effective strategies for the prevention or treatment (attenuation or reversion) of AMC, so patients remain at risk of developing serious cardiovascular events. Current treatment options for AMC focus on slowing the progression of arterial calcification and attempt to attenuate common risk factors such as hyperphosphatemia and hypercalcemia. These treatments, including phosphate binders, calcimimetics, and control of active vitamin D levels, lack efficacy since they target risk factors but do not directly inhibit calcification [9]. Therefore, effective therapeutic strategies directly blocking the development and/or progression of the calcification process are necessary. Development of such therapies would benefit from a more comprehensive understanding of cellular mechanisms and molecular pathways, leading to AMC. AMC development is known to be regulated by VSMCs. The latter transdifferentiate into cells with mineralizing capacity and lose their contractile phenotype as a result of circulating pathological stimuli such as high serum phosphate and glucose levels [10,11]. Even though ECs are the primary sensors of circulating pathological stimuli, there is a current paucity of evidence with respect to the involvement of ECs (and their interactions with VSMCs) in the AMC process.

In the present manuscript, we investigated cellular mechanism and molecular pathways underlying the AMC process by combining (i) an examination of cellular functionality (aortic reactivity) and interaction between VSMCs and ECs and (ii) quantitative proteomic analysis followed by functional signaling interpretation in/of calcified aortic tissue. The well-characterized warfarin murine model was used to induce AMC. Warfarin specifically inhibits the γ-glutamyl carboxylation of matrix Gla protein. The latter should normally become activated by carboxylation to prevent against calcification at binding calcium phosphate [12,13]. Warfarin, administered to CKD patients as an anti-coagulant, has also been shown to accelerate AMC in humans [14,15]. Cellular functionality was investigated using an ex-vivo, isometric organ bath set-up [16]. In short, we measured force and displacement generated by pharmacological VSMC mediated contraction and relaxation in aortic segments. Moreover, pharmacological induction of endothelial dysfunction allowed us to assess basal endothelial functionality in these aortic segments. Finally, the contribution of sarcoplasmic calcium to an elicited VSMC contraction could be estimated by performing the measurements in the absence of extracellular calcium.

## 2. Results

### 2.1. Warfarin Treatment Induces Aortic Media Calcification

Aortic calcium content was significantly increased in the aortic arch of warfarin treated mice compared with control mice (*p* = 0.0002) (Figure 1a). Calcified lesions were visually absent in the medial layer of control animals (Figure 1b) and present, as Von Kossa positive staining, in warfarin treated animals (Figure 1c).

### 2.2. Calcified Aortas Are Significantly More Sensitive to A1-Adrenergic Stimulated Vascular Smooth Muscle Cell Contraction

Cellular functionality of-and interaction between VSMCs and ECs was investigated using an ex-vivo, isometric organ bath setup. Compared to control animals, the concentration-response curve for PE of warfarin treated animals was significantly left shifted (*p* = 0.0018) (Figure 2a). Although aortic segments of warfarin and control animals reached the same maximal contraction (*p* = 0.7209) (Figure 2b), segments of warfarin administered mice were significantly more sensitive to α1-adrenergic stimulation by PE (*p* = 0.0207) (Figure 2c). The significance disappeared in the presence of ex vivo added L-NAME to the organ bath (*p* = 0.0650) (Figure 2d).

### 2.3. Vascular Smooth Muscle Cell Relaxation Remains Largely Unaffected in Calcified Aortas

ACh-mediated relaxation of VSMCs did not significantly differ between both warfarin and control animals (*p* = 0.4000) (Figure 3a). Both animal groups reached similar maximal relaxation (*p* = 0.2284) (Figure 3b) and were equally sensitive to the addition of ACh to the organ bath (*p* = 0.8518) (Figure 3c). To assess the ability of VSMC to relax independent of endothelial activity, DEANO (NO donor) was added to L-NAME containing organ baths. The dose response curve for increasing concentrations of DEANO was not significantly different between both groups (*p* = 0.8772) (Figure 3d). Equally, warfarin treatment did not induce significant differences for both maximal relaxation (*p* > 0.9999) (Figure 3e) and sensitivity (*p*= 0.7450) (Figure 3f).

### 2.4. Calcified Aortas Show Attenuated Phasic Contraction after A1-Adrenergic Stimulation in the Absence of Extracellular Calcium and Lower Contribution of Voltage-Gated Calcium Channels towards the Tonic Contraction

During the phasic contraction through IP_3_ mediated calcium release from the sarcoplasmic stores (in the absence of extracellular calcium), significantly less isometric force was generated in the warfarin group compared to controls (*p* = 0.0104) (Figure 4a). This difference in force completely disappeared after the addition of L-NAME to the organ bath (*p* = 0.5054) (Figure 4d). Tonic aortic contractions upon the addition 3.5 µM CaCl_2_ to the organ baths were not different between groups both in the absence (*p* = 0.1949) (Figure 4b) and presence of L-NAME in the organ bath (*p* = 0.3823) (Figure 4e). A lower (borderline insignificant *p* = 0.0650) percentage of relaxation inhibition was measured in the warfarin group after the addition of 35 µM of the VGCC blocker diltiazem (Figure 4c). This difference, which disappeared (*p* = 0.3037) after the addition of L-NAME to the organ bath (Figure 4f), indicates a larger contributory fraction of NSCCs towards the elicited tonic contraction prior to relaxation by diltiazem.

### 2.5. Protein Identification and Reactome Pathway Analysis Suggest a Potential Role for Endothelial Involvement in the Aortic Calcification Process in Mice

Significantly altered proteins of the acquired Maxquant proteomic datasets were compiled into a comprehensive log2-transformed ratio list of differentially expressed proteins (DEP list) that demonstrated a deviation of expression of greater than two standard deviations (plus or minus) from the background protein expression level (Table 1).

Notably, proteins associated with eNOS activity, which were significantly downregulated (versus the control group) in our DEP list, were Heat Shock Protein 90 Alpha Family Class A Member 1 (Hsp90aa1, −5.0), Voltage Dependent Anion Channel 1 (Vdac1, −3.8) and calcium/calmodulin-dependent protein kinase type II subunit beta (Camk2b, −4.3). Our DEP list includes proteins associated with endothelial to mesenchymal transition: the endothelial cell junction protein plakophilin-1 (PKP1, −4.3) and the mesenchymal protein fibronectin (Fn1, +5.8), which were significantly down- and upregulated in warfarin versus control mice, respectively. Moreover, numerous extracellular matrix proteins including Collagen (I) alpha-1chain (Col1a1, +5.8), Collagen (VI) alpha-1 and 3 chain (Col6A1, +5.3; Col6a2, +3.6; Col6A3, +4.6), Collagen (XVIII) alpha-1 chain (Col18a1, +10.2) were all upregulated after 8 weeks of warfarin administration. The latter, Col18a1, may give rise to endostatin by proteolytic splicing. In contrast, Collagen (XII) alpha-1 chain (Col12a1, −2.9) was significantly downregulated compared to the control group.

The dataset of differentially expressed proteins assessed via proteomics was verified by Western blotting analysis of aortic Vdac1 (Figure 5b). Downregulation of Vdac1 (Vdac1, −3.8) in the DEP list was confirmed by western blotting: significantly (*p* = 0.0286) less Vdac1 protein could be detected in the aortic tissue lysates of warfarin treated mice compared to control mice (Figure 5c).

Subsequently, a Reactome Pathway (https://reactome.org/) analysis (via NetworkAnalyst: https://www.networkanalyst.ca/, accessed on 28 July 2021) was performed from a Zero-Order human PPI network using a specific aortic protein database (Figure 5a, Table 2). Using this approach, we identified several protein interactions which could be clustered into pathways. A hybrid score (negative log10 of enrichment probability (*P*) multiplied by the enrichment ratio) was assigned for each specific pathway consisting of a subset of input proteins and bridging proteins. Most notably, proteins associated with vessel tonus and contractility were found to be differentially regulated in the calcified mouse aorta. Those were, proteins involved in eNOS activation (75.71 log_10_P*Enrichment) and muscle contraction (60.72 log_10_P*Enrichment). Given the impact of ectopic calcification of the aorta on both the cellular level and on the level of the cell surrounding matrix, proteins associated with DNA damage/cell senescence and extracellular matrix adaptations were also identified. Cellular pathways related to DNA damage/cell senescence which were found to be differentially regulated in the warfarin group were ‘Processing of DNA double strand breaks (DSBs)’ (188.11 log_10_P*Enrichment), ‘homologous recombination repair of DSBs’ (31.8 log_10_P*Enrichment) and ‘telomere maintenance’ (14.64 log_10_P*Enrichment). Lastly, related to the extracellular matrix, molecules associated with elastic fibers (79.02 log_10_P*Enrichment) and the elastic fiber formation (59.88 log_10_P*Enrichment) itself were affected in the warfarin group. Furthermore, the extracellular matrix was differently organized (27.2 log_10_P*Enrichment) and cell surface interactions at the vascular wall (19.28 log_10_P*Enrichment) were affected by the arterial calcification process.

## 3. Discussion

The role of ECs in the arterial calcification process might be overlooked.

In the present study, AMC was induced by administration of a warfarin containing diet. This model has been extensively described and widely used by others and is known to induce significant aortic calcifications in rodents [17]. In humans treated with warfarin, as an anti-coagulant, AMC is strengthened [14].

For a long time, AMC is thought to be a process exclusively regulated by VSMCs. However, the strategic positioning of the endothelium allows it to sense hemodynamic changes and respond appropriately to these changes. As such, the endothelium plays a crucial role in vascular homeostasis by maintaining the balance between endothelium-derived relaxing and contracting factors. In this respect its potential role in the pathophysiology of AMC might be largely overlooked. Nevertheless, in patient populations at risk for AMC such as type 2 diabetes patients and CKD patients the presence of endothelial dysfunction is well established [18,19,20]. The current study argues for a role of endothelial involvement in AMC in different ways.

To investigate interactions between VSMCs and ECs during the AMC process, we presented the assessment of aortic function via organ baths. We found that warfarin administered mice were significantly more sensitive to α1-adrenergic stimulation by PE. Release of basal NO generally protects against the α1-adrenergic VSMC contractions at higher tonus [21]. In patients, loss of this compensation mechanism, conceivably results in an increased sensitivity to circulating catecholamines, reinforcing arterial stiffness [22]. Basal NO release mainly restricts the contraction by PE through inhibiting the VSMC calcium influx via non-selective calcium channels. Pan-NOS inhibition by ex vivo added L-NAME to induce endothelial dysfunction is known to increase the relative contribution of NSCCs versus VGCCs towards the contraction by PE [23]. The increased sensitivity to PE administration could suggest that warfarin treatment negatively affects endothelial function. Furthermore, the warfarin group also reached a numerically lower mean maximal relaxation, in response to ACh. Remarkably, this form of stimulated NO release by ACh administration was not significantly attenuated, suggesting that only basal NO production was affected, while the receptor stimulated NO response was still intact in our AMC model. Compromised basal NO release together with an intact receptor-stimulated NO response was also seen earlier in organ baths of Fbn1^C1039G/+^ (Marfan model) [24], apolipoprotein E-deficient [25] isolated mouse aortic segments and arteries of stroke-prone spontaneously hypertensive rats [26]. Our study further exemplified the differences between basal and stimulated vasodilator NO activity in vascular diseases.

Adding PE to the organ bath in the absence of extracellular calcium resulted in an IP_3_-mediated, phasic contraction, that reflects the release of contractile calcium from the VSMC sarcoplasmic reticulum. The amplitude of this contraction was significantly lower in aortic segments from warfarin treated mice. Leloup and colleagues previously showed that a lack of basal eNOS activity and reduced NO release, lowers the calcium content of contractile PE-sensitive sarcoplasmic stores of the VSMC [27]. Again, our findings regarding IP_3_ mediated contraction, support the conclusion that, based on the organ bath experiments, warfarin treated animals show lower basal NO levels, compared to control animals.

The maximal force generated by the tonic contraction, elicited after restoring extracellular calcium, remained unaffected by warfarin treatment. Confirming that the aortic segments of the control and warfarin animals reached equal maximal contraction during the PE concentration-response experiment. Partial inhibition of the tonic contraction by the VGCC blocker diltiazem was less pronounced in warfarin treated mice. This difference was borderline insignificant. A possible explanation for this reduction in VGCC contribution towards the tonic contraction in warfarin treated animals, could be that the tonic contraction is predominantly due to calcium influx through NSCCs. In healthy mice, basal NO production is known to inhibit the influx of calcium through NSCCs [28]. Warfarin treatment in our experiment resulted in an enhanced contribution of NSCC towards the tonic contraction (since a smaller fraction of VGCC inhibition could be observed after the addition of diltiazem) again pointing into the direction of an attenuated NO availability. Finally, VSMC relaxation in the presence of the NO-donor DEANO remains normal, which reflects that VSMCs retained the ability to relax in case endothelial function would have been normal.

Reactome pathway analysis points into the direction of the endothelium contributing to warfarin-mediated calcification. Proteome analysis of our calcified aortic tissue confirmed the involvement of several pathological processes to be part of the AMC process. Not surprisingly, several proteins involved in collagen fibril production (Col1a1) and structural microfibril stabilization (Col6a1,2,3) were significantly upregulated in aortic tissue after warfarin administration. Remarkably, Col18a1 was significantly upregulated. Col18a1 may give rise to endostatin by proteolytic splicing. Serum endostatin levels are elevated in coronary artery disease, and have been reported as a sensitive marker for improved coronary artery calcification diagnosis and follow-up [29,30]. Interestingly, Col12a1 was the only protein from the collagen family, which was significantly downregulated in our model. Col12a1 functions as a bridging protein between collagen 1 fibril and the surrounding extracellular matrix [31]. These findings strongly suggest the arterial calcification in our model to be accompanied by extracellular matrix adaptations, a well-known phenomenon during AMC [32]. Furthermore, our Zero-order Human network analysis of the identified proteins also revealed the presence of a well-established DNA damage response in the warfarin group. The role of DNA damage has been extensively studied in the context of arterial calcification [33].

The proteomics dataset is a manifestation of all cell types present in mouse aortic tissue, therefore results considering the role that ECs play in AMC must be carefully interpreted. Nevertheless, the fact that our DEP encompasses a substantial number of proteins clearly related to endothelial function emphasizes the relevance of the endothelium, possibly contributing to the process of AMC. Our results strongly suggest that important chaperone molecules associated with eNOS might be negatively affected as a consequence of the warfarin administration. The direct interaction between voltage-dependent membrane channel Vdac1 and eNOS leads to a functional increase in eNOS activity [34]. Alvira et al. have previously demonstrated that the Vdac/eNOS interaction was significantly blunted in ovine pulmonary hypertensive artery ECs [35,36]. Additionally Hsp90, by directly interacting with eNOS [37], participates in the stabilization of the active eNOS form and preserves it from proteolytic degradation [38]. Decreased Hsp90/eNOS association is associated with significant impairment of NO release [39]. Interestingly, in our study, AMC was accompanied by a substantial down-regulation of both Vdac1 and Hsp90aa1 in abdominal aortic tissue. Hsp90aa1 was even the protein with the most pronounced downregulation in the DEP list (−5.0 times versus the control group), for Vdac1 we confirmed the 75% down-regulation in the DEP list by a significantly lower protein abundance by Western blot analysis. Also, proteomic analysis showed that Camk2, responsible for AMP-activated protein kinase (AMPK) phosphorylation and activation, was significantly downregulated in aortic tissue of warfarin mice. Loss of Camk2 expression, as seen in the DEP list, adversely affects the AMPK-eNOS pathway [40,41].

Reactome pathway analysis argues for an altered eNOS activity or a dysfunctional endothelium as a result of warfarin treatment Endothelial dysfunction is known as an initiator of endothelial to mesenchymal transition (EndMT) [42,43]. By undergoing EndMT, ECs could potentially function as an additional source of osteogenic progenitor cells in arterial calcification. In brief, EndMT involves loss of endothelial specific features and the subsequent acquisition of fibroblast-like characteristics with potential osteogenic-driven implications [44]. Upregulation of the mesenchymal gene *FN1*, encoding fibronectin, is a feature of the EndMT pathologic response [45]. Protein expression of fibronectin was significantly upregulated in the calcified aortas. Additionally, the observed repression of plakophilin 1, encoded by the *PKP1* gene, further reinforces EndMT by preventing the formation of cell-cell junctions between ECs, ultimately losing the endothelial barrier function [46]. Since we saw both an upregulation of fibronectin and a downregulation of plakophilin, one can speculate whether EndMT could be a part of the equation leading up to the deposition of media calcification in our mouse model. Overall, further research, specifically highlighting the role of ECs, is required to fully elucidate if indeed a dysfunctional endothelium, directly contributes to AMC. The latter being known in the field, to be a predominantly VSMC regulated process.

## 4. Materials and Methods

### 4.1. Animals

8-week-old, male DBA/2 mice (*n* = 16, Charles-River) were housed at the local animal facility of the University of Antwerp. They were kept in standard cages under 12–12 h light/dark cycles. All animals had free access to drinking water and chow ad libitum. The study was approved by the Ethical Committee of the University of Antwerp, and all experiments were performed according to the Guide for the Care and Use of Laboratory Animals 85–23 (1996). The mice were randomly assigned to two groups. The first group (*n* = 8) consisted of control mice receiving standard chow. The second group (*n* = 8) received a 3.0 mg/g warfarin- (vitamin K antagonist) 1.5% Vitamin K1 supplemented diet (Ssniff, Spezialdiäten GmbH, Soest, Germany) to induce AMC.

### 4.2. Quantification and Visualization of Arterial Media Calcification

All animals were sacrificed after 8 weeks. The aortic arch was harvested at sacrifice and briefly rinsed in 0.9% NaCl to remove excess blood and loose adherent tissue. Total calcium tissue content of the aortic arch was measured by flame atomic absorption spectrometry (FAAS). In brief, the aortic tissue was weighed on a precision balance and subsequently digested in 65% HNO_3_ at 60 °C overnight. Post volume adjustments, samples were diluted with 0.1% La(NO_3_)_3_ to eliminate chemical interference during the measurement. Total calcium content of the aortic tissue was expressed as mg calcium/g wet tissue. Part of the thoracic aorta was formalin fixed and paraffin embedded. Thoracic aortic sections were stained by the Von Kossa method to visualize calcified lesions [17].

### 4.3. Characterization of Aortic Reactivity at Cellular Level Using Ex Vivo Organ Bath Analysis

A section of the descending thoracic aorta was carefully removed and stripped of any extraneous adherent tissue. Starting at the diaphragm, the aorta was cut into segments of approximately 2 mm width using a calibrated stereomicroscope. Prior to the measurements, segments were immersed in Krebs Ringer (KR) solution (37 °C, 95% O_2_/5% CO_2_, pH 7.4) with (in mM): NaCl 118, KCl 4.7, CaCl_2_ 2.5, KH_2_PO_4_ 1.2, MgSO_4_ 1.2, NaHCO_3_ 25, CaEDTA 0.025, and glucose 11.1, and allowed to equilibrate for 60 min.

Isometric contractions and relaxations were measured in parallel on two different aortic segments per animal, by means of a Statham UC2 force transducer (Gould, Los Angeles, CA, USA). The aortic segments were stretched and stabilized to a preload of 25 mN. The pan NOS inhibitor L-NAME (300 µM) was added to every second organ bath (out of four) at the beginning of each experiment. A dose response curve for VSMCs contraction, using phenylephrine (PE, range of 3 nM–10 μM) (Sigma-Aldrich, Overijse, Belgium) was performed. Subsequently a dose response series of acetylcholine (ACh, range of 3 nM–10 μM) induced vessel relaxation was performed. In the presence of L-NAME, the NO donating molecule Diethylamine NONOate (DEANO, range of 0.3 nM–10 μM) was also employed in an additional vasodilatory dose response range to exclusively assess VSMC-mediated relaxation. Following all vessel stimulation procedures, extant active substances were removed by washing with KR.

Extracellular calcium was depleted from the KR solution by adding 1 mM ethylene glycol-bis (β-aminoethyl ether)-N,N,N′,N′-tetraacetic acid (EGTA, 0Ca KR) to the organ baths. Intracellular loading/storage of sarcoplasmic calcium, in the absence of extracellular calcium was quantified by measuring the isometric force generated by adding 2 µM PE to segments immersed in 0Ca KR (IP_3_-mediated calcium release from sarcoplasmic stores, phasic contraction). Next, extracellular calcium was restored, and a full, tonic contraction was subsequently elicited by addition of 3.5 µM CaCl_2_ (Still in the presence of 2 µM PE). Finally, diltiazem (35 µM) was added to the organ baths to quantify the % relaxation by inhibition of voltage gated calcium channels (%VGCC) and the contribution of non-selective calcium channels (%NSCC) towards the elicited contraction. A complete schematic overview of the used protocol is found in Figure A1.

### 4.4. Aortic Protein Isolation and Quantification Followed by Mass Spectrometric (MS)-Based Quantitative Proteomic Analysis

At sacrifice, a section (± 3–4 mm) of the abdominal aorta was immediately snap frozen in liquid nitrogen and later stored at −80 °C until further analysis. Aortic tissue was disrupted and sonicated in RIPA 4% SDS (150 mM NaCl, 50 mM Tris, 0.5% Sodium deoxycholate, 1% NP−40.4% sodium dodecyl sulphate). To prevent protein degradation by proteolytic and phospholytic enzymes, all procedures were conducted at 4 °C in the presence of PhosSTOP^TM^ phosphatase inhibitor cocktail tablets (Roche Diagnostics, Basel, Switzerland) and Complete^TM^ protease inhibitor cocktail tablets (Roche Diagnostics). Extracted protein lysate concentrations were determined using RC DC^TM^ Protein Assay (Bio-Rad). Samples for MS were prepared with the ProteoSpin™ on-column proteolytic digestion kit (Norgen Biotek). A nano-liquid chromatography (LC) column (Dionex ULTIMATE 3000) coupled online to a Q Exactive^TM^-Plus Orbitrap (ThermoScientific, Waltham, MA, USA) was used for the MS analysis. Peptides were loaded onto a 75 μm × 150 mm, 2 μm fused silica C18 capillary column, and mobile phase elution was performed using buffer A (0.05% formic acid in Milli-Q water) and buffer B (0.05% formic acid in 80% acetonitrile/Milli-Q water, Darmstadt, Germany). The peptides were eluted using a gradient from 5% buffer B to 95% buffer B over 120 min at a flow rate of 0.3 μL/min. The LC eluent was directed to an ESI source for Orbitrap analysis. The MS was set to perform data dependent acquisition in the positive ion mode for a selected mass range of 375–2000 *m*/*z* for quantitative expression difference at the MS1 (140,000 resolution) level followed by peptide backbone fragmentation with normalized collision energy of 28 eV, and identification at the MS2 level (17,500 resolution). The analysis was done via MaxQuant, a widely used software platform for the analysis of shotgun proteomics data available from Max Planck institute of biochemistry. Using MaxQuant allowed us to simultaneously identify and quantify the proteins (in a high-dimensionality manner) that are differentially up- or downregulated (at a *p* value of <0.05) in aortic samples of warfarin treated mice versus control mice. The software was connected to an Andromeda search engine. To assist in the bioinformatic interpretation of the protein lists generated, human orthologs were derived using the dbOrtho suite at bioDBnet (https://biodbnet-abcc.ncifcrf.gov/db/dbOrtho.php, accessed on 28 July 2021). Human orthologs were generated as the majority of effective informatic platforms employ human Gene Symbol identifiers. The significantly altered proteins of the acquired datasets were functionally annotated using the application of pathways from the Kyoto Encyclopedia of Genes (KEGG) database. In addition to this, a Reactome pathway analysis was performed upon a Zero-order human protein-protein interaction (PPI) network generated with the differentially expressed warfarin response data sets. The PPI Zero-Order network attempts to form the most coherent global network including the greatest number of input proteins (from the significantly regulated differentially expressed protein (DEP list)) through the inclusion of common connecting proteins from an aortic specific database (IMEx Consortium: https://www.imexconsortium.org/, accessed on 28 July 2021) between the input ‘seed’ proteins.

### 4.5. Western Blot

Specific protein expression level of Vdac1 (Voltage dependent anion selective channel 1) was assessed using abdominal aortic supernatant, derived as described in 2.4. The samples (volume needed to obtain 25 µg of proteins) were immersed and reduced in Laemmli buffer with 2.5% β-mercaptoethanol and heat-denatured for 5 min at 95 °C. Proteins were next separated by electrophoresis on a 10% Mini-PROTEAN TGC Stain-Free^TM^ Precast Gel (Bio-Rad, Hercules, CA, USA) at 200V being flanked by an unstained protein ladder (Precision Plus Protein Unstained Standards). Subsequently, the proteins were transferred to a PVDF membrane (30V, 70 min) and the membrane was blocked with 5% bovine serum albumin (BSA) in Tris-buffered saline with 0.1% Tween 20 (TBST) buffer for 1 h at room temperature. PVDF membranes were incubated overnight at 4 °C with the following primary antibody: Vdac1(1:1 000, ab14734: Abcam). Next, the membrane was washed extensively with TBST and incubated for 1 h at room temperature with horseradish peroxidase (HRP)-conjugated secondary rabbit anti-mouse (1:10,000, P0260) antibodies. All antibodies were diluted in 5% BSA in TBST. Finally, the membranes were washed again extensively, and proteins were detected via the enhanced chemiluminescent (ECL) system using the Clarity™ Western ECL Substrate. Protein levels were analyzed and quantified via total lane protein normalization in Image Lab (Bio-Rad).

### 4.6. Statistical Analysis

All results were expressed as mean ± SD, if not specified differently and were analyzed using Prism 8.4 (GraphPad Software). When testing the treatment factor in combination with other factors, a two-way ANOVA was used. To correct for multiple comparisons, a Sidak correction was applied. When comparing a single parameter between the treated and control group, a Mann–Whitney U test was used. All statistical tests are mentioned in the figure legends. Results were considered significant if *p* ≤ 0.05.

## 5. Conclusions

In summary, we performed an organ bath evaluation of isolated mouse aortic segments to approach arterial calcification from a multicellular point of view by looking at EC and VSMC interactions. A deteriorated basal endothelial nitric oxide function could be observed in calcified aortic rings. This finding is corroborated by the second part of the study, which investigated the underlying molecular mechanisms of AMC, as investigated by proteomics and reactome pathway analysis. The latter analysis puts forward an altered (reduced) eNOS activation as an underlying molecular mechanism of warfarin-induced AMC.

Certainly, more research on the precise contribution of the endothelium, and perhaps more specifically on the role of endothelial dysfunction, is needed to fully understand the pathophysiological mechanism of arterial calcification at the multicellular level.

## Figures and Tables

**Figure 1 ijms-22-11615-f001:**
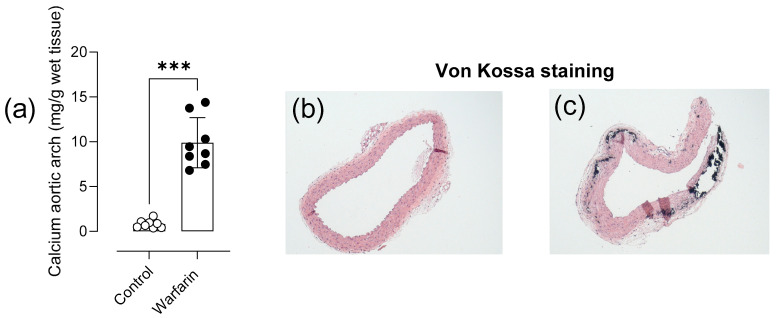
Warfarin treatment induces aortic media calcification. Quantification of the calcium content in the aortic arch (**a**) after 8 weeks of warfarin treatment. Von Kossa stained thoracic aortic section is shown for both control (**b**) and warfarin (**c**) treated mice (50× magnification). Calcified lesions are Von Kossa positive. (Control: *n* = 8, Warfarin: *n* = 8). Two-tailed Mann-Whitney U test was performed (**a**). Significance vs. control: *p* < 0.001: ***. Bar and error bars represent mean ± SD.

**Figure 2 ijms-22-11615-f002:**
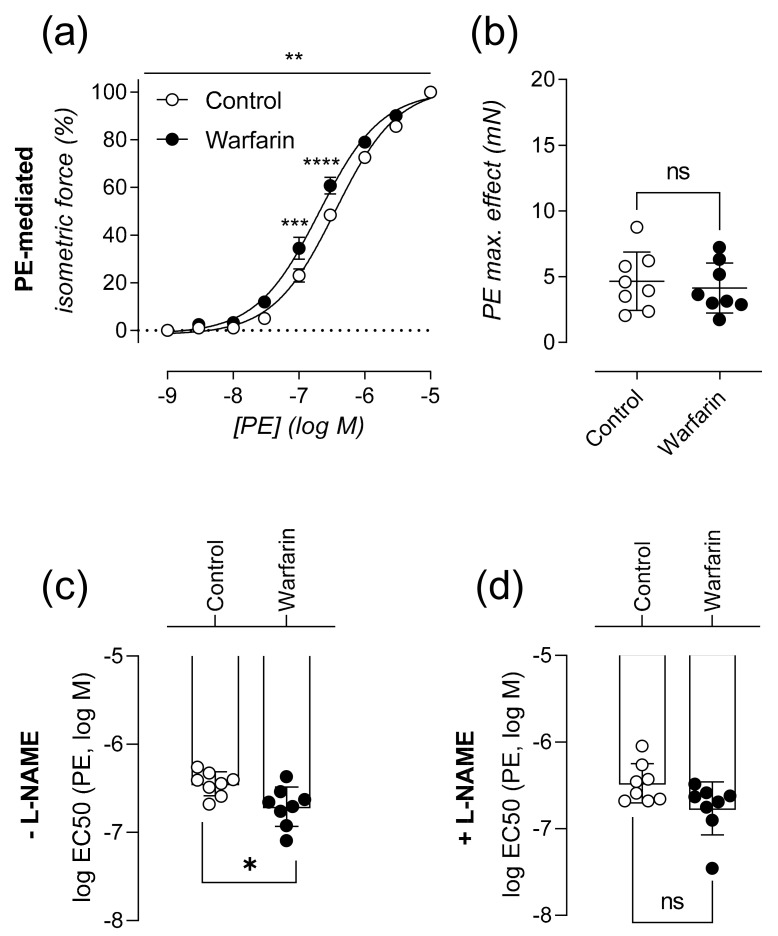
Calcified aortas are significantly more sensitive to α1-adrenergic stimulated vascular smooth muscle cell contraction. Relative isometric force is shown in relation to increasing concentrations of PE (%) in the absence L-NAME (**a**). Maximal PE mediated contraction (mN) (**b**) and sensitivity to PE, expressed as logEC50 (**c**) in the absence of L-NAME are shown. LogEC50 in the presence of L-NAME (**d**) (Control: *n* = 8, Warfarin: *n* = 8). Two-way ANOVA with Sidak multiple comparison correction was performed for the dose response curve. A significant effect of warfarin treatment (*p* < 0.01) was found (**a**). Two-tailed Mann-Whitney U tests were used to assess statistical significance between two groups (**b**–**d**). Significance summary or vs. control: *p* ≤ 0.05: ns, *p* ≤ 0.05: *, *p* ≤ 0.01: **, *p* ≤ 0.001: ***, *p* ≤ 0.0001: ****. Lines/bars and error bars represent mean ± SEM (**a**) and mean ± SD (**b**–**d**).

**Figure 3 ijms-22-11615-f003:**
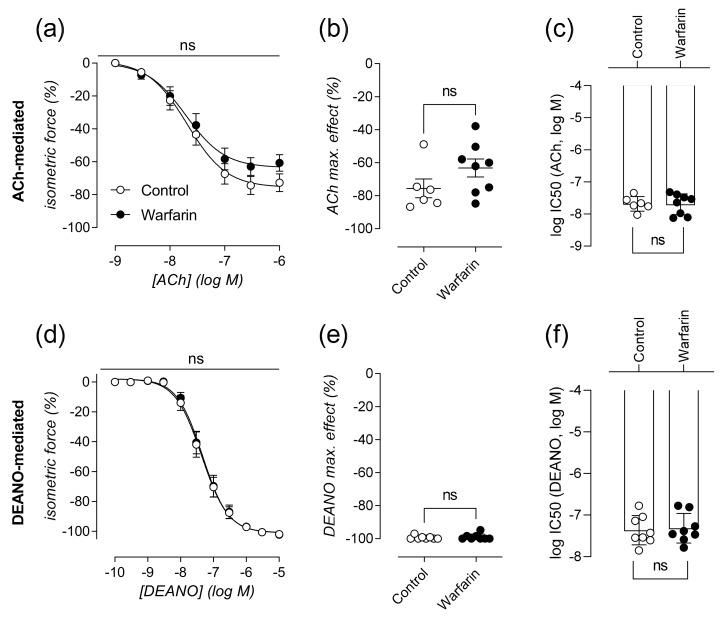
Vascular smooth muscle cell relaxation remains largely unaffected in calcified aortas. Relative isometric force is shown in relation to increasing concentrations of ACh (%) (**a**) leading to relaxation of the aortic segment. Maximal ACh-mediated relaxation (%) (**b**) and sensitivity to ACh (logIC50) (**c**) are shown. In the presence of L-NAME, the relative isometric force is shown in relation to increasing concentrations of DEANO (**d**). Maximal DEANO mediated relaxation (%) (**e**) and sensitivity to DEANO (logIC50) (**f**) are shown. ACh:(Control: *n* = 6, Warfarin: *n* = 8), DEANO:(Control: *n* = 8, Warfarin: *n* = 8). Two-way ANOVA with Sidak correction was performed for the dose response curves. (**a**,**d**). Two-tailed Mann-Whitney U test was used to assess statistical significance between two groups (**b**–**f**). Significance summary or vs. control: *p* > 0.05: ns. Lines/bars and error bars represent mean ± SEM (**a**,**d**) and mean ± SD (**b**–**f**).

**Figure 4 ijms-22-11615-f004:**
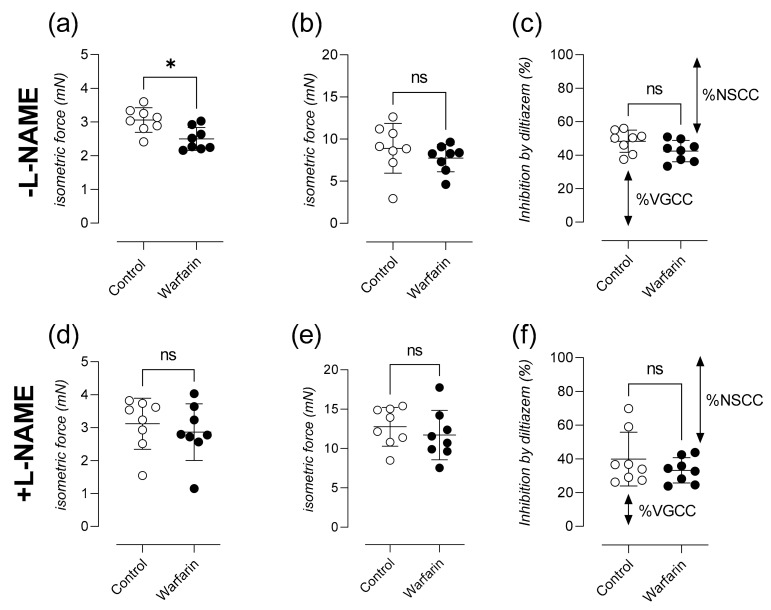
Calcified aortas show attenuated phasic contraction after α1-adrenergic stimulation in the absence of extracellular calcium and lowered contribution of voltage-gated calcium channels towards the tonic contraction. Maximal force generated during the phasic contraction by 2 µM PE in the absence of extracellular calcium (0Ca KR) is shown (**a**). Next, extracellular calcium was restored by addition of 3.5 mM CaCl_2_ and a tonic contraction was generated (**b**). Finally, the tonic contraction was partly inhibited using 35 µM diltiazem (**c**). All measurements were performed in parallel with pan-nitric oxide synthase inhibition after the addition of L-NAME to the organ bath (**d**–**f**). VGCC and NSCC fractions are visualized by arrows. (Control: *n* = 8, Warfarin: *n* = 8). Mann-Whitney U test was used to assess statistical significance between two groups (**a**–**f**) Significance vs. control: *p* > 0.05: ns, *p* ≤ 0.05: *. Lines and error bars represent mean ± SD.

**Figure 5 ijms-22-11615-f005:**
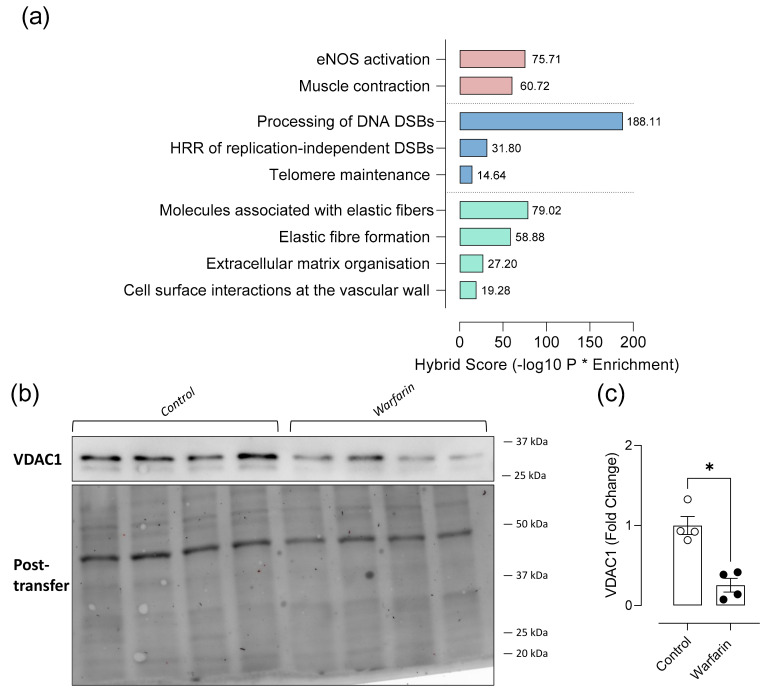
Protein identification and Reactome pathway analysis suggest a potential role for endothelial involvement in the aortic calcification process in mice. Reactome pathway analysis was performed from a Zero-order human PPI network using a specific aortic protein database. A hybrid score (negative log10 of enrichment probability multiplied by the enrichment ratio) was assigned for each specific pathway consisting of a subset of input proteins and bridging proteins. (**a**). Expression of aortic Vdac1 was visualized using Western blotting (**b**) and total lane protein normalization (Post-transfer) was done using ImageLab (Biorad) software (**c**) (Control: *n* = 4, Warfarin: *n* = 4). Two-tailed Mann Whitney U test was used to assess statistical significance between two groups (**c**). Significance vs control: *p* ≤ 0.05: *. Lines and error bars represent mean ± SD.

**Table 1 ijms-22-11615-t001:** List of differentially expressed proteins (DEP list) analyzed by MaxQuant. Fold change protein expression of warfarin vs. control aortic tissue is shown in green (upregulation) or red (downregulation).

Protein Name	Gene	Warf/Control Expression
Collagen alpha-1(XVIII) chain;Endostatin	Col18a1	10.20285161
Myosin light polypeptide 6	Myl6	9.305566367
Actin, cytoplasmic 1	Actb	7.13605868
Hemoglobin subunit beta-2	Hbb-b2	6.283635977
Protein S100-A10	S100a10	6.1668153
Lumican	Lum	6.089201997
Actin, alpha cardiac muscle 1	Actc1	5.900030588
Fibronectin	Fn1	5.846997258
Collagen alpha-1(I) chain	Col1a1	5.84074422
Transgelin	Tagln	5.83338011
Pyruvate kinase PKM	Pkm	5.630031596
Histone H4	Hist1h4a	5.563012253
Myosin regulatory light polypeptide 9	Myl9	5.437436649
Collagen alpha-1(VI) chain	Col6a1	5.343617272
Fibulin-5	Fbln5	5.332078817
Ubiquitin-60S ribosomal protein L40	Uba52	5.265733513
Peptidyl-prolyl cis-trans isomerase A	Ppia	5.150343559
Histone H3	H3f3a	5.127585135
Tropomyosin beta chain	Tpm2	4.986908606
Periostin	Postn	4.663392316
Collagen. type VI, alpha 3	Col6a3	4.656731287
Integrin beta-1	Itgb1	4.636192833
Histone H2A.J	H2afj	4.632914586
Annexin A1	Anxa1	4.629622078
Mimecan	Ogn	4.568776349
ATP synthase subunit beta, mitochondrial	Atp5b	4.560486285
Biglycan	Bgn	4.474612835
78 kDa glucose-regulated protein	Hspa5	4.411751218
Histone H1.4	Hist1h1e	4.260793326
Protein disulfide-isomerase A3	Pdia3	3.959701928
Transgelin-2	Tagln2	3.913522447
Phosphoglycerate mutase 1	Pgam1	3.792118568
Tubulin beta-4B chain	Tubb4b	3.781716637
Actin, aortic smooth muscle	Acta2	3.728309917
Collagen alpha-2(VI) chain	Col6a2	3.626587586
Heat shock cognate 71 kDa protein	Hspa8	3.572501281
Elongation factor 1-alpha 1	Eef1a1	3.543392106
Myelin basic protein	Mbp	3.51233397
Histone H2A	Hist1h2al	3.491398962
Hemoglobin subunit beta-1	Hbb-b1	3.459772146
Glyceraldehyde-3-phosphate dehydrogenase	Gapdh	3.453170438
Desmoplakin	Dsp	3.428314207
Integrin alpha-8;Integrin alpha-8 heavy chain;Integrin alpha-8 light chain	Itga8	3.205454819
Prolargin	Prelp	3.154404844
AHNAK nucleoprotein (desmoyokin)	Ahnak	3.152708981
Latent-transforming growth factor beta-binding protein 4	Ltbp4	3.061532269
Heat shock protein beta-1	Hspb1	3.042730966
Microfibril-associated glycoprotein 4	Mfap4	2.990339431
Myosin-11	Myh11	2.970878687
60 kDa heat shock protein, mitochondrial	Hspd1	2.942801207
ATP synthase subunit alpha, mitochondrial;ATP synthase subunit alpha	Atp5a1	2.866588069
Lamin-B1	Lmnb1	2.755784918
Peroxiredoxin-1	Prdx1	2.741488404
Mitochondrial pyruvate carrier 2	Mpc2	−2.776854365
Sorcin	Sri	−2.778554479
Fumarate hydratase. mitochondrial	Fh	−2.78790387
Isocitrate dehydrogenase [NAD] subunit alpha, mitochondrial	Idh3a	−2.827601808
Dystroglycan;Alpha-dystroglycan;Beta-dystroglycan	Dag1	−2.843683644
RNA-binding protein FUS	Fus	−2.855025156
Adiponectin	Adipoq	−2.870786473
Fibrillin-1	Fbn1	−2.881231656
Caveolin;Caveolin-1	Cav1	−2.881356055
Collagen alpha-1(XII) chain	Col12a1	−2.900177418
Non-histone chromosomal protein HMG-17	Hmgn2	−2.901948849
Cell cycle exit and neuronal differentiation protein 1	Cend1	−2.921660364
Tropomyosin alpha-1 chain	Tpm1	−2.939741546
Dystrophin	Dmd	−2.942721979
60S ribosomal protein L11	Rpl11	−3.014421333
40S ribosomal protein S8	Rps8	−3.035124354
3-ketoacyl-CoA thiolase, mitochondrial	Acaa2	−3.048082514
Neurofilament medium polypeptide	Nefm	−3.07662492
Aspartate aminotransferase, cytoplasmic	Got1	−3.080449284
Annexin;Annexin A4	Anxa4	−3.105836221
Alcohol dehydrogenase [NADP(+)]	Akr1a1	−3.106485937
Ras-related protein Rab-6A	Rab6a	−3.115239165
Isocitrate dehydrogenase [NADP], mitochondrial	Idh2	−3.115706253
2,4-dienoyl-CoA reductase. mitochondrial	Decr1	−3.118324745
Sodium/potassium-transporting ATPase subunit beta-2	Atp1b2	−3.127998158
Glycogen phosphorylase, brain form	Pygb	−3.135272346
Fibrinogen gamma chain	Fgg	−3.191670257
Heterogeneous nuclear ribonucleoprotein A3	Hnrnpa3	−3.199867691
60S ribosomal protein L22	Rpl22	−3.227769432
40S ribosomal protein S13	Rps13	−3.236784722
Serine protease HTRA1	Htra1	−3.321747522
EGF-containing fibulin-like extracellular matrix protein 2	Efemp2	−3.336806766
Microtubule-associated protein 1B	Map1b	−3.344559981
Lamina-associated polypeptide 2, isoforms beta/delta/epsilon/gamma	Tmpo	−3.357538495
ABI gene family, member 3 (NESH)-binding protein	Abi3bp	−3.384864137
Heterogeneous nuclear ribonucleoprotein U	Hnrnpu	−3.406417617
Poly(rC)-binding protein 2	Pcbp2	−3.425399964
Myotrophin	Mtpn	−3.466273576
ADP-ribosylation factor 4	Arf4	−3.492602394
60S ribosomal protein L30	Rpl30	−3.536454437
Heat shock 70 kDa protein 1B	Hspa1b	−3.541089505
Transforming growth factor beta-1-induced transcript 1 protein	Tgfb1i1	−3.54939462
RNA binding motif protein, X-linked-like-1	Rbmxl1	−3.600255079
LIM domain-binding protein 3	Ldb3	−3.600647311
Rab GDP dissociation inhibitor alpha	Gdi1	−3.77816993
Dystrobrevin;Dystrobrevin alpha	Dtna	−3.839881707
Voltage-dependent anion-selective channel protein 1	Vdac1	−3.896705419
Lamin-B2	Lmnb2	−3.906904033
Elongation factor 1-gamma	Eef1g	−3.908570365
Splicing factor, proline- and glutamine-rich	Sfpq	−3.924620008
L-lactate dehydrogenase;L-lactate dehydrogenase A chain	Ldha	−4.038149336
F-box only protein 50	Nccrp1	−4.090311808
Proteasome subunit alpha type-6	Psma6	−4.176758295
Actin-related protein 2/3 complex subunit 1B	Arpc1b	−4.19220217
Basal cell adhesion molecule	Bcam	−4.260099113
Calcium/calmodulin-dependent protein kinase type II subunit beta	Camk2b	−4.262372799
Alpha-2-HS-glycoprotein	Ahsg	−4.270275838
Plakophilin-1	Pkp1	−4.274045522
C-type lectin domain family 11 member A	Clec11a	−4.307868231
Chymase	Cma1	−4.480664848
Phosphatidylinositol-binding clathrin assembly protein	Picalm	−4.528845203
Motile sperm domain-containing protein 2	Mospd2	−4.592705873
Synaptosomal-associated protein 25	Snap25	−4.60289817
Calponin	Cnn2	−4.732552032
Protein NDRG1	Ndrg1	−4.741722435
Heat shock protein HSP 90-alpha	Hsp90aa1	−5.014769768

**Table 2 ijms-22-11615-t002:** Reactome pathway analysis performed upon a Zero-order human PPI network generated with the differentially expressed warfarin response data sets. False Discovery rate (FDR), enrichment ratio (Enrichment), negative log10 of enrichment probability multiplied by the enrichment ratio (Hybrid Score).

Reactome Pathway	Total	Expected	Hits	*p* Value	FDR	Enrichment	Hybrid Score
Processing of DNA double-strand break ends	3	0.0764	3	0.0000162	0.00134	39.26701571	188.1080492
Molecules associated with elastic fibres	38	0.968	10	2.24 × 10^−8^	0.0000104	10.33057851	79.02636345
eNOS activation	9	0.229	4	0.0000463	0.00265	17.46724891	75.7103757
Muscle contraction	52	1.32	11	5.16 × 10^−8^	0.0000145	8.333333333	60.72791915
Elastic fibre formation	45	1.15	10	0.00000013	0.0000273	8.695652174	59.87875346
Homologous recombination repair of replication-independent double-strand breaks	16	0.407	4	0.000581	0.0209	9.828009828	31.80170877
Extracellular matrix organization	157	4	17	0.000000397	0.0000618	4.25	27.20514035
Cell surface interactions at the vascular wall	99	2.52	11	0.0000382	0.00233	4.365079365	19.28464405
Telomere Maintenance	72	1.83	8	0.000447	0.017	4.371584699	14.64346438

## Data Availability

Data will be available upon request of the reviewers/readers.

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
