# Peer review of "Endothelial Contribution to Warfarin-Induced Arterial Media Calcification in Mice"

_ijms, 2021, doi:10.3390/ijms222111615_

Round 1

Reviewer 1 Report

The authors aimed to investigates cellular functioning and molecular pathways underlying AMC, respectively by, an ex-vivo isometric organ bath set-up  to explore the interaction between VSMCs and ECs and quantitative proteomics followed by functional pathway interpretation. AMC development, which was induced in mice by dietary warfarin  administration, was proved by positive Von Kossa staining and a significantly increased calcium  content in the aorta compared to that of control mice.

The study covers some issues that have been overlooked in other similar topics. The structure of the manuscript appears adequate and well divided in the sub-paragraphs. Moreover, the study is easy to follow, but some issues should be improved. The manuscript needs moderate grammar correction. Please also check typos thorough the text. Conclusion Section: This paragraph is too long: please take in account a general revision to eliminate redundant sentences and to add some "take-home message".

Author Response

We thank the reviewer for reading the manuscript and providing us with useful feedback. We agree the Conclusion section was too lengthy and therefore shortened this paragraph. The conclusion paragraph now contains the main findings (take-home messages) and ends by suggesting the need to advance our knowledge regarding the contribution of the endothelium towards AMC pathophysiology. Furthermore, we proofread the manuscript again to check for grammar errors/typos.

Reviewer 2 Report

In the current manuscript the authors have used warfarin induced arterial calcification in mice as a model to draw a relationship between he calcification process and not only ASM but also endothelial cells. However there are a few concerns to be address 

  1. Protein quantification by Western blot requires an internal control such as GAPDH or Beta actin to ensure equal loading of protein across the blots. Provide a simultaneous blot for internal control and the respective VDAC-1 band intensities can be normalized to the internal control.  
  1. The proteomics has been done on the entire aorta from the ex vivo culture. So the dataset is a manifestation of all cell types of aorta. It is hard to distinguish the role of endothelial cells in calcification process with the reactome. The data lacks the rigor in drawing the role of endothelial cells in calcification. The proteomics data can be explained better in the results and discussion 

Author Response

We thank the reviewer for reading the manuscript and providing us with useful feedback.

As mentioned in the manuscript (line 442-443), we used the stain-free blotting technique (Bio-rad, for reference https://www.bio-rad.com/webroot/web/pdf/lsr/literature/Bulletin_6351.pdf), which enabled visualization of the post-transfer total protein amount in each lane. This allows us to quantify and normalize the chemiluminescent signal of our target protein (VDAC1) with the post-transfer total protein (instead of only 1 internal control protein) image as a reference. To be fully transparent, we have included the full analysis report (ImageLab) in the resubmission (as pdf). The post-transfer total protein image was added to figure 5.

We agree that the proteomics dataset entails proteins from different cell types and that some sort of disclaimer would be appropriate in the manuscript. Therefore, we adapted and moved a paragraph prior to starting the discussion about the endothelial involvement seen in the Reactome analysis results. We believe this improves the readability of the discussion. In order to avoid textual repeats in results and discussion sections, we skipped redundant sentences from the description of the proteomic data in the result section.  
